# IMAGINATION POLICY: Using Generative Point Cloud Models for Learning Manipulation Policies

**Haojie Huang[1], Karl Schmeckpeper[†2], Dian Wang[†1], Ondrej Biza[†1,2], Yaoyao Qian[‡1],**
**Haotian Liu[‡3], Mingxi Jia[‡4], Robert Platt[1,2], Robin Walters[1]**

†, ‡ Equal Contribution, [1]Northeastern University, Boston, MA 02115, USA
[2]Boston Dynamics AI Institute, [3]Worcester Polytechnic Institute, [4]Brown University
{huang.haoj; r.platt; r.walters}@northeastern.edu
https://haojhuang.github.io/imagine_page/

**Abstract:** Humans can imagine goal states during planning and perform actions to match those goals. In this work, we propose IMAGINATION POLICY, a novel multi-task key-frame policy network for solving high-precision pick and place tasks. Instead of learning actions directly, IMAGINATION POLICY generates point clouds to imagine desired states which are then translated to actions using rigid action estimation. This transforms action inference into a local generative task. We leverage pick and place symmetries underlying the tasks in the generation process and achieve extremely high sample efficiency and generalizability to unseen configurations. Finally, we demonstrate state-of-the-art performance across various tasks on the RLbench benchmark compared with several strong baselines and validate our approach on a real robot.

**Keywords:** Manipulation policy learning, Generative model, Geometric learning

## 1 Introduction

Humans can look at a scene and imagine how it would look with the objects in it rearranged. For example, given a flower and a bottle on the table, we can imagine the flower placed in the bottle. Using this mental picture, we can then manipulate the objects to match the imagined scene. However, most robotic policy learning algorithms shortcut this process and map observations directly to actions ($SE(3)$ poses or displacements) [1, 2, 3, 4, 5, 6]. These approaches lose important information about the desired geometry of the scene and are therefore less sample efficient and less precise than they might be.

Inspired by how humans solve tasks, we propose IMAGINATION POLICY which takes two point clouds as input and generates a new point cloud combining the inputs into a desirable configuration using a conditional point flow model (see Figure 1). Given the generated point cloud, we use point cloud registration methods to match the observed input point clouds with the "imagined" scene. This gives rigid body transformations which can be used to command a robot arm to manipulate the objects. IMAGINATION POLICY consists of two generative processes, each of which uses the above method. As shown in Figure 1a, the *pick generator* generates the points of the object positioned relative to the gripper point cloud. The *place generator* generates a pair of objects rearranged together as shown in Figure 1b. Compared to directly generating actions, this adds many degrees of freedom to the generative process which aids optimization and sensitivity to geometric interactions.

IMAGINATION POLICY addresses two key challenges in current multitask manipulation policy learning: high precision manipulation and sample efficient learning. Methods like PerAct [1], RVT [2] and Diffuser Actor [3] struggle to learn high precision manipulation policies such as those required to solve the RLBench [7] tasks *Plug-Charger* and *Insert-Knife*. IMAGINATION POLICY outperforms on these tasks by enabling the model to reason about detailed geometric interaction,

8th Conference on Robot Learning (CoRL 2024), Munich, Germany.

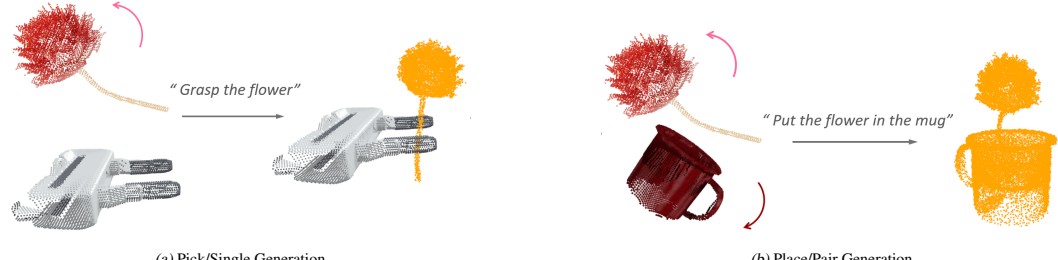

(a) Pick/Single Generation          (b) Place/Pair Generation

*Figure 1.* Illustration of pick generation and place generation. The pick generator generates the points of the object to be picked conditioned on the gripper point cloud. The place generator generates two new objects repositioned together. The generated points are colored in orange.

such as how different parts of an object's surface should be displaced, which in turn facilitates precise reasoning about the movement of a tool tip to align with a mating surface. Our method also excels in sample efficiency, the ability to learn good policies with relatively few expert demonstrations. Because IMAGINATION POLICY reasons about the desired relative configuration of two objects, it can more easily incorporate symmetries of two object systems, called bi-equivariance, into the model. This significantly improves the sample efficiency of the model. While previous work [8, 9, 10, 11] has also used this kind of bi-equivariant structure to improve sample efficiency, our method is the first to apply the idea outside of the pick-place setting in the more general *keyframe multitask* setting. While IMAGINATION POLICY can still solve pick-place tasks, it can also solve more general manipulation tasks like *Plug-Charger*, *Insert-Knife*, and *Open-Microwave*.

Our contributions in this paper are as follows. 1) We are the first to propose a generative point cloud model to estimate desired rigid body motion in a keyframe manipulation setting. 2) We show how to implement $SE(3)$ bi-equivariant constraints in this setting. 3) We demonstrate that the resulting method, IMAGINATION POLICY, achieves state-of-the-art performance on several RLBench tasks against several strong baselines.

## 2 Related Work

**Point Cloud Generation.** Previous works have explored point cloud generation using VAEs [12, 13] and GANs [14, 15]. Recently, score-based denoising models and normalizing flows [16, 17, 18, 19, 20, 21] have demonstrated the power and flexibility to generate high-quality point clouds. For example, Zhou et al. [18] proposed a probabilistic diffusion approach for unconditional point cloud generation. Luo and Hu [19] and Vahdat et al. [20] formulated conditional point cloud diffusion. PSF [22] achieved fast point cloud generation with rectified flow [21]. Ours, however, generates pick and place point clouds conditioned on the observation that can be used to estimate a rigid action to command the robot arm.

**Point Clouds in 3D Pick-and-place Manipulation.** Point clouds provide a flexible, geometric representation to encode object shapes and poses. In terms of pick-and-place manipulation, Simeonov et al. [23] used SE(3)-invariant point features to encode descriptor fields enabling sample efficient policy learning. Simeonov et al. [4] extended Diffusion Policy [24] to work with point cloud observations and to learn multimodal actions. Pan et al. [25] propose TAX-POSE which is closely related to our method. Pan et al. [25] used two segmented point clouds as input and directly output a new point cloud using a weighted summary of target points together with residual predictions. Concurrently, Eisner et al. [26] adopted TAX-POSE [25] with relative distance inferred by a kernel method. However, it was designed to output the new point cloud directly in one step without penalty for the generated results. Our method, instead, uses generative models to predict the movement of each point iteratively with a velocity model. Moreover, they are limited to single-task training, only work in one-step pick-and-place settings, and thus cannot be applied to complex tasks without predefined prior actions. Recently, Shridhar et al. [1], Goyal et al. [2], and Ke et al. [3] showed impressive multi-task capabilities with transformer-based architectures. However, these methods require hundreds of expert demonstrations and cannot successfully learn high-precision tasks. In contrast, our

method leverages bi-equivariant symmetry and amortizes the action prediction across multiple tasks. As a result, it can solve high-precision tasks with few demonstrations.

**Symmetry in Robot Learning.** Robotic tasks defined in 3D Euclidean space are invariant to translations, rotations, and reflections which redefine the coordinate frame but do not otherwise alter the task. Recent advancements in equivariant modeling, such as those discussed by [27, 28, 29, 30], offer a convenient approach to capturing these symmetries in robotics. Zhu et al. [31] and Huang et al. [32] utilized equivariant models to enforce pick symmetries for grasp learning. Yang et al. [33] proposed an equivariant policy for deformable and articulated object manipulation on top of pre-trained equivariant visual representation. Other works [23, 34, 8, 35, 9, 10, 11, 36, 37, 38, 39] leverage symmetries in pick and place and achieve high sample efficiency. However, they are limited to single-task pick-and-place equivariance. As a result, they cannot be directly applied to the *Plug-Charger* and *Insert-Knife* tasks without a pre-place action. Our proposed method, however, can achieve bi-equivariance in the *key-frame* and multi-task setting. In addition, we employ equivariant action inference using an invariant point cloud generating process, which is different from previous methods.

# 3 Method

**Problem Statement.** Consider a dataset $\mathcal{D}$ with samples of the form $(P_a, P_b, T_a, T_b, P_{ab}, \ell)$ where $P_a \in \mathbb{R}^{n \times 3}$ and $P_b \in \mathbb{R}^{m \times 3}$ are point clouds that represent two segmented objects, $P_{ab} \in \mathbb{R}^{(n+m) \times 3}$ represents the two objects at the desired configuration described by the language instruction $\ell$, and $T_a \in \mathbb{R}^{4 \times 4}$ and $T_b \in \mathbb{R}^{4 \times 4}$ are two rigid transformations in SE(3) represented in homogeneous coordinates that transform $P_a$ and $P_b$ into the desired configuration, i.e., $P_{ab} = T_a \cdot P_a \cup T_b \cdot P_b$. As shown in Figure 1, for the pick, $(P_a, P_b)$ indicates the gripper and the object to pick (the flower). For the place, it represents the placement (the mug) and the object to arrange (the flower). In either the pick or place setting our goal is model the policy function $f \colon (P_a, P_b, \ell) \mapsto a$ which outputs the gripper movement $a \in \mathrm{SE}(3)$. We consider placing to include the pre-place action and the place action.[1]

**Imagination Policy.** We factor action inference into two parts, point cloud generation (Figure 2ab) and transformation inference (Figure 2c). In the first part, we train a generative model which, when conditioned on $\ell$, generates a new coordinate for each point of $P_a$ and $P_b$ to approximate $P_{ab}$, i.e., $f_{\mathrm{gen}} \colon (P_a, P_b, \ell) \mapsto (\hat{P}_a, \hat{P}_b)$ where $\hat{P}_a \cup \hat{P}_b \approx P_{ab}$. In the second part, we estimate two transformations $\hat{T}_a$ from $P_a$ to $\hat{P}_a$, and $\hat{T}_b$ from $P_b$ to $\hat{P}_b$ using singular value decomposition (SVD) [40]. Then, the pick action of the gripper can be calculated as $a_{\mathrm{pick}} = (\hat{T}_b)^{-1} \hat{T}_a$ and the pre-place and place action can be estimated as $a_{\mathrm{place}} = (\hat{T}_a)^{-1} \hat{T}_b$.

## 3.1 Pair Generation for Place

We first explain how the above method works in the place setting $f_{\mathrm{place}} \colon (P_a, P_b, \ell) \mapsto a_{\mathrm{place}}$. The generative model $f_{\mathrm{gen}}$ has two sequential parts, a point cloud feature encoder (Figure 2a) and a conditional generator (Figure 2b). Then, we calculate the transformation $a_{\mathrm{place}}$ from the generated points (Figure 2c). Finally, we prove a condition for when the full method is bi-equivariant.

**Encoding Point Feature.** Given $P_a = \{p_a^i\}_{i=1}^n$ and $P_b = \{p_b^j\}_{j=1}^m$, we first compute a feature at each point using two point cloud encoders $\phi_a$ and $\phi_b$. The encoder $\phi_a$ takes the XYZ coordinate and RGB color of all points of $P_a$ as input and outputs pointwise features $\{f_a^i\}_{i=1}^n$. Similarly, $\phi_b \colon P_b \mapsto \{f_b^j\}_{j=1}^m$, which shares an architecture but has separate parameters.

**Generating Points.** The combined point cloud $P_{ab}$ is generated conditioned on the point features $F_a = \{f_a^i\}_{i=1}^n$ and $F_b = \{f_b^j\}_{j=1}^m$ using a modified version of Point Straight Flow [22]. This is a generative flow model where, at inference time, samples $X_1$ are taken by flowing over a vector field parameterized by a neural network $v_\theta$. Initial conditions are given by $X_0 = X_0^{P_a} \cup X_0^{P_b} = \{x_0^k\}_{k=1}^{n+m}$

---

[1]The pre-place action is the prerequisite to perform the place action. An example is shown in Figure 4c.

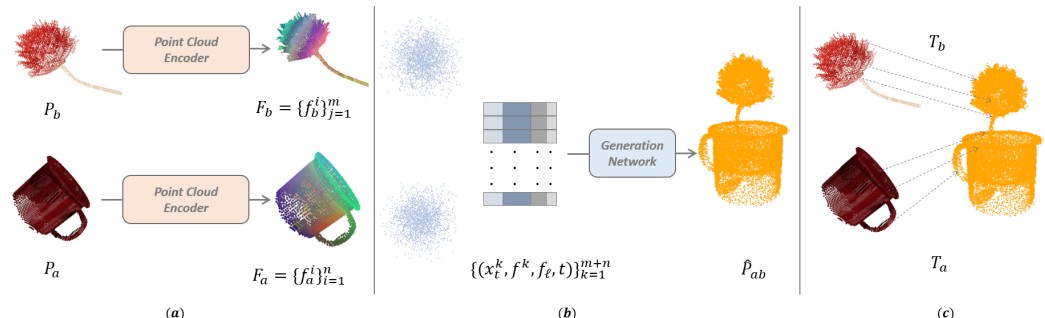

Figure 2. Architecture of IMAGINATION POLICY. (a). Encoding the observed point features as $F_a$ and $F_b$. (b). Conditional pair generation of the place scene from random Gaussian noise. $x_t^k$ illustrates the $k$-th noise at time step $t$ with the point feature $f^k$ and $f_\ell$ is the language feature. (c). Estimating the rigid transformation ($T_a$ and $T_b$) from the observed point cloud to the generation using correspondence.

where $X_0^{P_a} \in \mathbb{R}^{n \times 3}$ and $X_0^{P_b} \in \mathbb{R}^{m \times 3}$ which are sampled from a scaled Gaussian. The network $v_\theta$ defines the vector field for the ODE:

$$\mathrm{d}X_t = v_\theta(X_t, F_a, F_b, f_\ell, t) \, \mathrm{d}t, \quad t \in [0, 1] \tag{1}$$

where $X_t$ is the intermediate point cloud states at time t and $f_\ell$ is the encoded language feature of the language description $\ell$ from CLIP [41]. To solve the ODE, we iteratively update $X_{t+\Delta t} = X_t + v_\theta(Z_t)\Delta t$ for $\frac{1}{\Delta t}$ steps. The model is trained by setting the optimal direction at any time $t$ as $P_{ab} - X_0$ which provides the objective,

$$\min_\theta E(||v_\theta(X_t, F_a, F_b, f_\ell, t) - (P_{ab} - X_0)||^2) \tag{2}$$

where $X_t = tP_{ab} + (1 - t)X_0$. Intuitively, this sets $\mathrm{d}X_t$ to be the drift force needed to move $X_t$ to $P_{ab}$. Specifically, for a single point $p^k \in P_a \cup P_b$, we sample a noise $x_0^k$ and a time step $t$ to calculate the intermediate point

$$x_t^k = t(Tp^k) + (1 - t)x_0, \ \ T = T_\alpha \text{ if } p^k \in P_\alpha \text{ where } \alpha \in \{a, b\} \tag{3}$$

The generator input is $p^k = (x_t^k, f^k, f_\ell, t)$. We then optimize $\theta$ with respect to the loss function defined in Equation 2.

**Estimating the Action.** Given two sets of points $P_a \cup P_b = (p_a^1, p_a^2, \cdots, p_a^n) \cup (p_b^1, p_b^2, \cdots, p_b^m)$ and their corresponding target positions $P_{ab} = (T_a p_a^1, T_a p_a^2, \cdots, T_a p_a^n) \cup (T_b p_b^1, T_b p_b^2, \cdots, T_b p_b^m)$, we can recover the rigid transformations $T_a$ and $T_b$ using SVD [40]. However, the output $\hat{P}_a \cup \hat{P}_b$ of the generator $f_{\mathrm{gen}}$ is not constrained to be given by rigid transforms of the original two point clouds. Each point may move independently by transformation $T_\alpha^i$ such that $\hat{P}_a \cup \hat{P}_b = (T_a^1 p_a^1, T_a^2 p_a^2, \cdots, T_a^n p_a^n) \cup (T_b^1 p_b^1, T_b^2 p_b^2, \cdots, T_b^m p_b^m)$. We can still use SVD to estimate the best fitting $\hat{T}_a$ between $(P_a, \hat{P}_a)$ as well as $\hat{T}_b$ between $(P_b, \hat{P}_b)$. Assuming $P_b$ represents the object to be arranged and $P_a$ represents the placement, as shown in Figure 2, the pre-place or place action can be calculated as $a_{\mathrm{place}} = (\hat{T}_a)^{-1}\hat{T}_b$.

**Realizing Bi-equivariance.** As noted in prior work [8, 9, 10], place actions that transform an object B with respect to another object A are bi-equivariant. That is, independent transformations of object B with $g_b \in \mathrm{SE}(3)$ and object A with $g_a \in \mathrm{SE}(3)$ result in a change ($a'_{\mathrm{place}} = g_a a_{\mathrm{place}} g_b^{-1}$) to complete the rearrangement at the new configuration. Leveraging bi-equivariant symmetries can generalize learned place knowledge to different configurations and improve sample efficiency. Our placement model is constrained to be bi-equivariant, with invariant generation during training.

**Proposition 1.** *Assuming rotation-invariant Gaussian noise $X_0$, if the encoded point feature $F_a$ and $F_b$ are invariant to rotations then $f_{\mathrm{place}}$ is bi-equivariant*

$$f_{\mathrm{place}}(g_a \cdot P_a, g_b \cdot P_b) = g_a f_{\mathrm{place}}(P_a, P_b)g_b^{-1}$$

*for all pairs of rotations $(g_a, g_b) \in \mathrm{SO}(3) \times \mathrm{SO}(3)$.*

*Proof.* If $X_0 = \{x_0^k\}_{k=1}^{n+m}$ and $F_a \cup F_b = \{f^k\}_{k=i}^{n+m}$ are rotation-invariant, the intermediate point states $X_t = tP_{ab} + (1 - t)X_0$ are rotation invariant with a fixed $P_{ab}$. Since all inputs to $v_\theta$ are

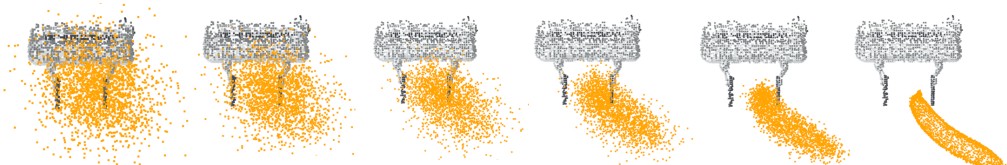

*Figure 3.* Trajectory of the pick generation process ("grasp the banana by the crown"). Unlike the place generation, our pick generation is conditioned on the canonicalized gripper point cloud. The generated point cloud at each timestep is colored in orange.

invariant and the output always approaches $P_{ab}$, we have

$$f_{\text{gen}}(g_a \cdot P_a, g_b \cdot P_b) = f_{\text{gen}}(P_a, P_b) \tag{4}$$

With the same generated points $P_{ab}$, the estimated transformation from rotated observation $g_a P_a = (g_a p_a^1, g_a p_a^2, \cdots, g_a p_a^n)$ to $\hat{P}_a$ is $\hat{T}_a g_a^{-1}$. Similarly, the estimated transformation from $g_b P_b$ to $\hat{P}_b$ is $\hat{T}_b g_b^{-1}$. Then, the new place action $a'_{\text{place}}$ can be calculated as $a'_{\text{place}} = (\hat{T}_a g_a^{-1})^{-1} \hat{T}_b g_b^{-1} = g_a \hat{T}_a^{-1} \hat{T}_b g_b^{-1} = g_a a_{\text{place}} g_b^{-1}$, which satisfies bi-equivariance . $\qquad\square$

## 3.2   Single Generation for Pick

Our pick network $f_{\text{pick}} \colon (P_a, P_b) \mapsto a_{\text{pick}}$ has a similar design to $f_{\text{place}}$. In this setting, $P_a$ are the points of the gripper and $P_b$ are the points of the object to pick. The function $f_{\text{pick}}$ differs from $f_{\text{place}}$ in that we only generate the new points for $P_b$ conditioned on $P_a$. Figure 3 illustrates the generation process of grasping the banana by the crown.

Since the pose and the shape of the gripper are always known, we fix $P_a$ in a canonical pose, sample only $X_0^{P_b}$ from a Gaussian distribution, and construct $X_0 = P_a \cup X_0^{p_b}$. We set the target $P_{ab}$ as the union of the canonicalized gripper with the point cloud $P_b$ posed so it is held by the gripper, i.e., $P_{ab} = P_a \cup T_b P_b$. We only use $P_b$ to calculate the loss:

$$\min_\theta E(||v_\theta(X_t, F_a, F_b, f_\ell, t) - (T_b P_b - X_0^{P_b})||^2) \tag{5}$$

After estimating $\hat{T}_b$ from $(P_b, \hat{P}_b)$, the pick action is calculated as $a_{\text{pick}} = \hat{T}_b^{-1}$.

**Proposition 2.** *Assuming rotation-invariant Gaussian noise $X_0^{P_a}$, $f_{\text{pick}}$ is equivariant to rotations on the pick target if the encoded point feature $F_a$ and $F_b$ are rotation-invariant: $f_{\text{pick}}(P_a, g_b \cdot P_b) = g_b f_{\text{pick}}(P_a, P_b)$.*

Specifically, if there is a rotation $g_b$ acting on $P_b$, the generated points $\hat{P}_b$ are the same as those without rotation. The estimated transformation from $g_b P_b$ to $\hat{P}_b$ is $\hat{T}_b g_b^{-1}$ and the new pick action can be calculated as $a'_{\text{pick}} = (\hat{T}_b g_b^{-1})^{-1} = g_b \hat{T}_b^{-1}$, which realizes the desired equivariance property.

## 4   Experiments

**Model Architecture Details.** The generative models $f_{\text{pick}}$ and $f_{\text{place}}$ share the same architecture. Each has two point cloud encoders and a generation network. We select PVCNN [42] as the backbone of our point encoders, which output a 64-dimension feature for each point. We use a pretrained CLIP-ViT32 [41] model as our language encoder and project the language embedding to a 32-dimension vector with a linear layer. The time step $t$ is encoded as a 32-dimension positional embedding. We also encode a binary mask that indicates if the point belongs to $P_a$ or $P_b$ as a 32-dimension positional embedding. As a result, the generator input of a point is a 163-dimension vector. We adopt PSF [22] as our generator backbone. Both $f_{\text{pick}}$ and $f_{\text{place}}$ are trained end-to-end with the MSE loss defined in Equation 2 and Equation 5. We use the Adam optimizer with an initial learning rate of $10^{-4}$. Training takes 7 hours to converge with 200k training steps on a single RTX-4090 graphic card. During inference, we randomly sample $X_0$ from a Gaussian distribution and integrate over $v_\theta$ with 1000 steps to generate $P_{ab}$ and calculate the action. Generating one batch takes 20 seconds.

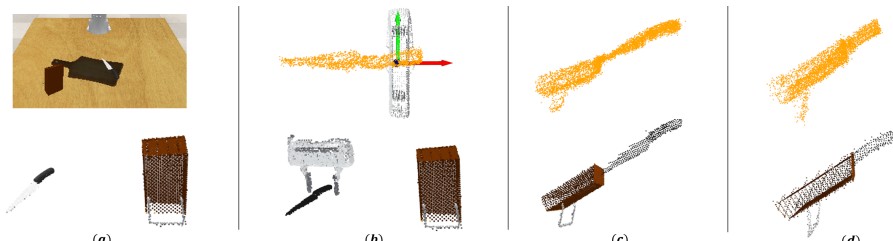

*Figure 4.* Illustration of the keyframe pipeline of IMAGINATION POLICY on *Insert-Knife*: (a) the RGB-D image captured by the front camera and the segmented point clouds, (b) pick generation, (c) preplace generation, and (d) place generation. The top row shows the generated points with orange color and the bottom row demonstrates the configurations of pick, preplace, and place with the calculated rigid transformations.

All point clouds $P_a$ and $P_b$ in our experiments are captured by RGB-D cameras instead of directly sampling from the ground truth mesh. We first center the point cloud and then downsample by selecting at most one point in each cell of a 4mm voxel grid. We further randomly subsample or duplicate to get 2048 points for $P_a$ and $P_b$. To get rotation-invariant generation, we apply extensive SO(3) data augmentation to $P_a$ and $P_b$ during training, i.e., an SO(3) rotation is sampled uniformly at each training step. This enforces $f_{\text{gen}}(g_a P_a, g_b P_b) = f_{\text{gen}}(P_a, P_b)$, which leads to the desired symmetry properties from Proposition 1. We found the results slightly outperform the equivariant point encoder of Vector Neurons [28], as shown in Table 3. We hypothesize that VN's expressivity is not as strong as that of PVCNN.

## 4.1  3D Key-frame Pick and Place

We conduct our primary experiments on six tasks shown in Figure 5 from RLbench [7] and compare it with three strong multi-task baselines [1, 2, 3].

**3D Task Description.** We choose the six difficult tasks from James et al. [7] to test our proposed method. ***Phone-on-Base:*** The agent must pick up the phone and plug it onto the phone base correctly. ***Stack-Wine***: This task consists of grabbing the wine bottle and putting it on the wooden rack at one of three specified locations. ***Put-Plate***: The agent is asked to pick up the plate and insert it between the red spokes in the colored dish rack. The colors of other spokes are randomly generated from the full set of 19 color instances. ***Put-Roll***: This consists of grasping the toilet roll and sliding the roll onto its stand. This task requires high precision. ***Plug-Charger:*** The agent must pick up the charger and plug it into the power supply on the wall. Thus is also a high-precision task. ***Insert-Knife:*** This task requires picking up the knife from the chopping board and sliding it into its slot in the knife block. The different 3D tasks are shown graphically in Figure 5. Object poses are randomly sampled at the beginning of each episode and the agent must generalize to novel poses.

**Baselines.** Our method is compared against three strong baselines: ***PerAct*** [1] is the state-of-the-art multi-task behavior cloning agent that tokenizes the voxel grids together with a language description of the task and learns a language-conditioned policy with Perceiver Transformer [43]. ***RVT*** [2] projects the 3D observation onto five orthographic images and uses the dense feature map of each image to generate 3D actions. ***3D Diffuser Actor*** [3] is a variation of Diffusion Policy [24] that denoises noisy actions conditioned on point cloud features. Comparison with this baseline tests the importance of point cloud generation since this baseline generates actions directly. ***RPDiff*** [4] consumes segmented $P_a$ and $P_b$ and denoises the relative pose iteratively. To make a fair comparison, we adapt [4] to a multi-task policy. See Appendix 6.4 for our implementation details. NDFs [23] and its variation [34] are not included since they require per-object pretraining.

**Settings.** All methods are trained as multi-task models. There are four cameras (front, right shoulder, left shoulder, hand) pointing toward the workspace. For our method, we formulate the action sequence as (pick, preplace, place), as shown in Figure 4. Specifically, our method generates the pick action with $f_{\text{pick}}$, and the preplace and place action with $f_{\text{place}}$ simultaneously. We use the ground truth mask to segment $P_a$ and $P_b$ for RPDiff [4] and our method, as shown in Figure 4a.

**Training and Metrics.** We train our method with 1, 5, or 10 demonstrations and train the baselines with 10 demonstrations. All methods are evaluated on 25 unseen configurations and each

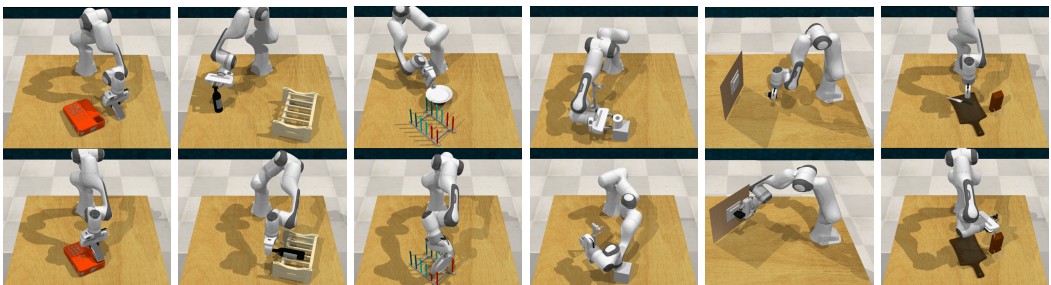

*Figure 5.* 3D pick-place tasks from RLBench [7]. From left to right the tasks are: *Phone-on-Base, Stack-Wine, Put-Plate, Put-Roll, Plug-Charger*, and *Insert-Knife*. The top row shows the initial scene and the bottom row shows the completion state.

| Model | # demos | phone-on-base | stack-wine | put-plate | put-roll | plug-charger | insert-knife |
|---|---|---|---|---|---|---|---|
| IMAGINATION POLICY (ours) | 1 | 4.00 | 2.67 | 1.33 | 2.78 | 0 | 0 |
| IMAGINATION POLICY (ours) | 5 | 78.67 | **97.33** | 0 | 1.39 | 24.00 | 38.67 |
| IMAGINATION POLICY (ours) | 10 | **90.67** | 97.33 | 34.67 | **23.61** | **26.67** | **42.67** |
| RVT [2] | 10 | 56.00 | 18.67 | **53.33** | 0 | 0 | 8.00 |
| PerAct [1] | 10 | 66.67 | 5.33 | 12.00 | 0 | 0 | 0 |
| 3D Diffusor Actor [3] | 10 | 29.33 | 26.67 | 12.00 | 0 | 0 | 0 |
| RPDiff [4] | 10 | 62.67 | 32.00 | 5.33 | 0 | 0 | 2.67 |
| Key-Frame Expert | | 98.67 | 100 | 74.6 | 56 | 72 | 90.6 |

*Table 1.* Performance comparisons on RL benchmark. Success rate (%) on 25 tests when using 1,5, or 10 demonstration episodes for training. Results are averaged over 3 runs. Even with only 5 demos, our method can outperform existing baselines by a significant margin.

evaluation is averaged over 3 evaluation seeds. We report the mean success rate of each method in Table 1. Since some tasks are very complex, to measure the effects of path planning, we also report the performance of the key-frame formulation used by our method with poses from the expert demonstrations (Key-Frame Expert) as an upper bound on performance.

**Results.** We report the results of all methods in Table 1. Several conclusions can be drawn from Table 1: 1) IMAGINATION POLICY significantly outperforms all baselines trained with 10 demos on all the tasks except *Put-Plate*. It can also achieve over $90\%$ success rates in *Phone-on-Base* and *Stack-Wine*. 2) For tasks with a high-precision requirement, e.g., *Plug-Charger*, *Insert-Knife* and *Put-Roll*, IMAGINATION POLICY has a relatively high success rate while all the baselines fail to learn a good policy. 3) IMAGINATION POLICY achieves better sample efficiency and demonstrates few-shot learning performance. With one or five demonstrations, it sometimes outperforms the baselines trained with 10 demonstrations, e.g., IMAGINATION POLICY achieves a $97.2\%$ success rate on *Stack-Wine* trained with 5 demos while the best baseline can only achieve $32\%$. We believe this sample efficiency is due to the models equivariance which allows it exploit the symmetry inherent in the generation task. In the end, our method underperforms one baseline in *Put-Plate*. We hypothesize that the object in this task is symmetric and is hard to encode with distinguishable point features, which might result in wrong correspondences when estimating the rigid transformations. Since many complex manipulation tasks can be decomposed as a sequence of single pick and place, we illustrate that our method can address long-horizon tasks in Appendix 6.2.

### 4.2 Real Robot Experiment

We validated IMAGINATION POLICY on a physical robot. We trained a multi-task agent from scratch on 3 tasks using a total of just 30 demonstrations. There was no use of the simulated data or pretraining in this experiment – all demonstrations were performed on the real robot.

**Settings.** The experiment was performed on a UR5 robot with a Robotiq-85 end effector, as shown in Figure 6a. The workspace was a $48\text{cm} \times 48\text{cm}$ region on a table. There were three RealSense 455 cameras mounted pointing toward the workspace. We split the workspace into two parts to place the object and the placement. The segmented point cloud was directly obtained by cropping the workspace accordingly. To collect the demonstrations, we released the UR5 brakes to push the arm physically and record data of the form (initial observation, pick pose, preplace pose, place pose). The combined point cloud $P_{ab}$ was constructed with segmented points and the poses. During testing, we used MoveIt as our path planner to execute the action sequentially.

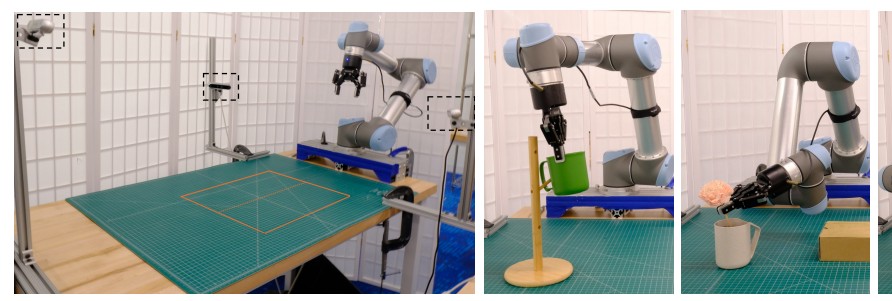

| (a) Workspace Settings | (b) Mug-Tree | (c) Plug-Flower | (d) Pour-Ball |

*Figure 6.* Settings and tasks of real-world experiments.

| Task | # demos | # pick completions | # place completions | # completions / # trials | success rate |
|---|---|---|---|---|---|
| Mug-Tree | 10 | 15/15 (100%) | 12/15 (80.0%) | 12 /15 | 80.0% |
| Plug-Flower | 10 | 15/15 (100%) | 14/15 (93.3%) | 14/15 | 93.3% |
| Pour-Ball | 10 | 14/15 (93.3%) | 14/14 (100%) | 14/15 | 93.3% |

*Table 2.* Performance on real-world experiments.

**Tasks.** We evaluate IMAGINATION POLICY on three pick and place tasks, as shown in Figure 6bcd. ***Mug-Tree:*** The robot needs to pick up the mug and place it on the mug holder. ***Plug-Flower:*** This task consists of picking up the flower and plugging it into the mug. ***Pouring-Ball:*** The agent is asked to grasp the small blue cup and pour the ball into the big green cup.

**Results.** We collected 10 human demonstrations of each task. Our model was trained for 200k SGD steps with the same settings as the simulated experiments. We evaluated 15 unseen configurations of each task. The results are reported in Table 2. Visualizations of the captured observation and the generated actions are shown in Appendix 6.5. Videos can be found in supplementary materials. Our failures are mainly caused by the distortion of observations and motion planning errors. For example, the handle of the green mug in *Mug-Tree* task might disappear due to sensor noise and calibration, which results in a place failure.

## 5 Conclusion

In this work, we propose IMAGINATION POLICY, a multi-task model for manipulation pick and place problems. It utilizes point cloud generation for key-frame manipulation policy learning by reasoning about the geometric configuration of the goal state. This process amortizes action prediction during generation by estimating the drift force of each point. We also analyze the key-frame equivariance of the task and implement it in the model by learning rotation-invariant point features. IMAGINATION POLICY demonstrates high sample efficiency and superior performance on six challenging RLbench tasks against several strong baselines. Finally, we demonstrate that the method can effectively be used to learn manipulation policies on a physical robot. We test our design choices using an ablation study on a multimodal pick-part dataset in Appendix 6.1.

One limitation of the formulation in this paper is that it relies on segmented point clouds. We believe state-of-the-art segmentation models [44, 45] are sufficient to provide high-quality masks. Additionally, our generation process takes 20 seconds with 1000 steps to finish point cloud generation. Fortunately, a large number of works have studied a range of methods for improving the inference speed of diffusion models [46, 47, 48, 49, 50, 51, 22]. We leave applying these existing techniques to future work. Moreover, this paper mainly focuses on rigid-object manipulation. We add one experiment of articulated object manipulation in Appendix 6.3. Our method might also work well for deformable objects. We will explore articulated objects and deformable objects in future work. Lastly, this paper assumes a fixed one-to-one correspondence between points in the object point clouds and generated point clouds. However, our pipeline of generation and pose estimation proposed here does not strictly require this. Specifically, one can generate point clouds without a correspondence and then train a point cloud registration model to estimate the transformations.

**Acknowledgments**

This project were supported in part by NSF 1750649, NSF 2107256, NSF 2314182, NSF 2134178, NSF 2409351, and NASA 80NSSC19K1474. Dian Wang was also funded by the JPMorgan Chase PhD fellowship. We would like to thank Jung Yeon Park and Nichols Crawford Taylor for their helpful discussions.

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

# 6 Appendix

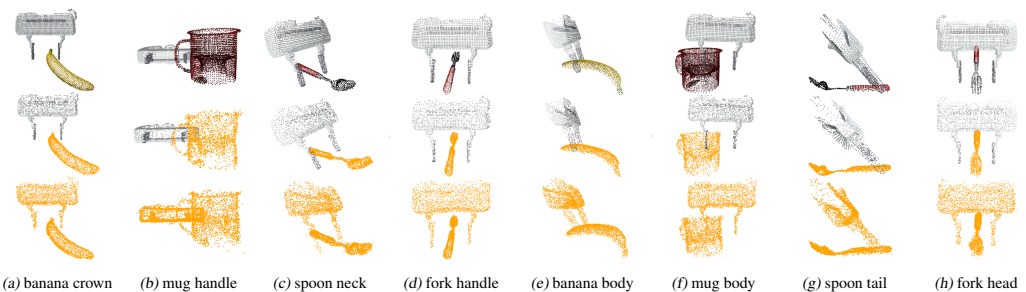

(a) banana crown     (b) mug handle     (c) spoon neck     (d) fork handle     (e) banana body     (f) mug body     (g) spoon tail     (h) fork head

*Figure* 7. Visualization of pick generation and place generation. Top row: multimodal pick-part training labels for different objects. Mid row: single generation for pick . Bottom row: pair generation for place. The generated point cloud is colored in orange. Note the model takes the randomly rotated and downsampled point cloud as input.

| max \| mean \| min | Pick/Single Generation | | | | | | Place/Pair Generation | | | | | |
|---|---|---|---|---|---|---|---|---|---|---|---|---|
| | rot error (°) | | | trans error (cm) | | | rot error (°) | | | trans error (cm) | | |
| Imagination Policy | 0.16 | 2.02 | 4.82 | 0.43 | 0.94 | 1.72 | 0.36 | 2.34 | 7.12 | 0.55 | 1.05 | 1.76 |
| w/o downsample | 0.47 | 7.77 | 39.55 | 0.43 | 0.96 | 1.77 | 0.29 | 9.28 | 28.94 | 0.65 | 1.19 | 1.85 |
| w/o color | 0.76 | 4.41 | 16.18 | 0.18 | 0.87 | 2.41 | 0.39 | 5.05 | 21.22 | 0.40 | 0.97 | 1.76 |
| w/o augmentation | 17.45 | 125.26 | 179.27 | 0.44 | 1.50 | 6.53 | 49.24 | 130.60 | 178.59 | 1.31 | 14.46 | 30.80 |
| PointNet Encoder | 0.42 | 3.30 | 10.09 | 0.26 | 0.88 | 1.56 | 0.96 | 4.82 | 14.31 | 0.33 | 0.98 | 1.74 |
| Pretrained VN Encoder | 0.75 | 5.24 | 34.06 | 0.26 | 0.80 | 1.56 | 0.92 | 6.01 | 20.68 | 0.44 | 1.04 | 2.13 |

*Table 3.* Ablation Results. We report the minimum, mean, and maximum error for single generation and pair generation over 100 runs with randomly rotated and sampled input.

## 6.1 Ablation Study

**Multimodal Pick-part Dataset.** To quantitatively measure the point cloud generation results and the equivariance of IMAGINATION POLICY, we create a small pick-part dataset using four YCB objects [52] (banana, mug, spoon and fork). We load each object in the Pybullet simulator [53] and use three cameras to get the RGB-D images to extract the point cloud. Each object is assigned with two different expert grasps with corresponding language instructions, e.g., "grasp the mug by the handle", "grasp the mug by its body", as shown in Figure 7.

**Training and Metrics.** We trained a single pick generation model to generate all the objects conditioned on the canonicalized gripper points and language descriptions. We also train a place generation model to generate both the gripper point cloud and the object point cloud. To evaluate the pick generation results, we randomly rotate and randomly downsample the object point cloud ($P_b$) to make a starting pose unseen during training. We calculate the translation error and rotation error between the estimated grasp pose and the ground truth grasp pose. Note that rotating $P_b$ results in a change of the ground truth pick pose. To evaluate the place generation results, we randomly rotate and downsample the gripper ($P_a$) as well as the object ($P_b$) to make a scene unseen during training and calculate the translation error and rotation error between estimated transformation $\hat{T}_a^{-1}\hat{T}_b$ and the ground truth pose. Note that rotating either $P_a$ or $P_b$ changes the relative ground truth transformation. We report the minimum, mean and maximum error over 100 runs in Table 3. We also show visualizations of the generated point clouds in orange in Figure 7.

**Results.** Table 3 includes 6 variations of our proposed methods. Several findings can be concluded from Table 3: (1) As shown in the first row, IMAGINATION POLICY can learn the multimodal distribution and is equivariant. It realizes around $2^{\circ} \sim 3^{\circ}$ average rotation error and 1cm translation error with different configurations of $P_a$ and $P_b$; (2) Without downsampling or color information, the rotation error slightly increases; (3) Without data augmentation in training, the performance decreases dramatically since the model cannot learn rotation-invariant features. (4) Compared with

the results in the last two rows, the PVCNN-based point cloud encoder outperforms PointNet [54] and the pre-trained equivariant point cloud encoder from NDF [23]. Note that the pre-trained point cloud encoder consumes enormous 3D point clouds from ShapeNet [55] and makes use of Vector Neuron [28] which is guaranteed to output the rotation invariant feature. We hypothesize that the architecture of Vector Neuron [28] and the standard representation limit its expressivity.

## 6.2 Task with Longer Horizon

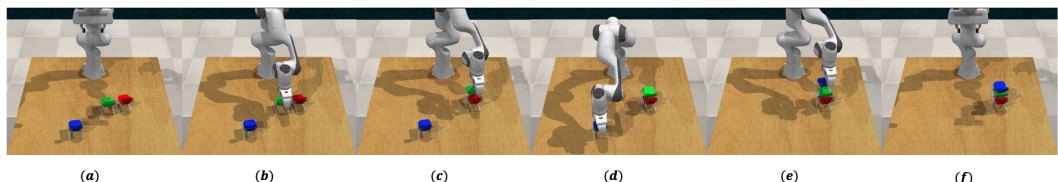

*Figure 8.* Illustration of Stack-three-Chairs. From left to right: (a). initial observation, (b). pick the green chair, (c). place the green chair, (d). pick the blue chair, (e). place the blue chair, (f). complete state.

| Task | # demos | first pick-place success rate | second pick-place success rate | overall success rate | oracle performance |
|---|---|---|---|---|---|
| Stack-Three-Chairs | 10 | 70.66% ± 2.61% | 68.00%±4.52% | 48.05% ± 3.71% | 88.0%± 4.52% |

*Table 4.* Performance on Stack-Three-Chairs. Success rate (%) on 25 tests using 10 demonstration episodes for training. Results are averaged over 3 runs. For each test, the poses of the three chairs are randomly sampled with a different seed from the training data.

Many challenging robotic manipulation problems can be viewed through the lens of a single pick and place operation. We test IMAGINATION POLICY on the task with a longer horizon. ***Stack-three-Chairs*** requires picking the other two chairs and stacking them on top of the base (red) chairs following the RGB order. This is a high-precision task requiring the agent to correctly manage a sequence of pick and place. Even trained with 10 demos, our method can achieve 70.66% success rates in the first pick-place execution and maintain a similar performance (68.00%) for the second pick-place execution. Detailed results are reported in Table 4. It demonstrates that IMAGINATION POLICY can address long-horizon tasks.

## 6.3 Task with Articulated Object

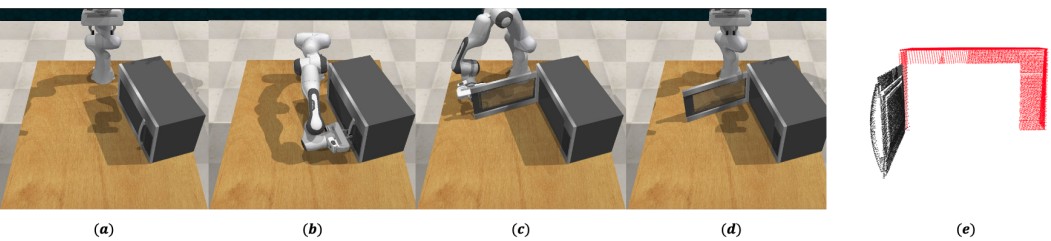

*Figure 9.* Illustration of Open-Microwave. From left to right: (a). initial observation, (b). pick the handle of the microwave, (c). open the door of the microwave, (d). final complete state, (e). segmentations of the door with handle (black color) and frame (red color).

| Task | # demos for training | success rate | oracle performance |
|---|---|---|---|
| Open-Microwave | 10 | 69.33 %± 5.22% | 90.66% ±% 2.61% |

*Table 5.* Performance on Open-Microwave. Success rate (%) on 25 tests using 10 demonstration episodes for training. Results are averaged over 3 runs. For each test, the pose of the microwave is randomly sampled with a different seed from the training data.

Articulated objects are special cases in manipulation since they are linked with several movable parts. We test IMAGINATION POLICY on Open-Microwave task to illustrate the potential of our method of addressing articulated object manipulation. Specifically, we segment the two movable parts of the microwave, as shown in Figure 9e, the door with the handle (black color) and the frame

(red color). The task consists of grasping the handle of the microwave and opening the door. In our settings, the grasping is to infer the relative pose between the gripper and the door; the opening is to predict the relative pose between the door and the frame. Even trained with 10 demonstrations, our method can achieve 69.33% success rate, as reported in Table 5. We found that most failure cases are due to the motion planning error and the collision between the door and the gripper when the gripper is closing. More complex articulated objects can also be manipulated by predicting the relative poses between links and we leave it as the future work. Overall, it demonstrates that IMAGINATION POLICY can address articulated object manipulation.

## 6.4 Baseline Details

The closest standard methods evaluated on RLBench [7] are PerAct [1], RVT [2], and 3D Diffuser Actor [3]. These are multi-task key-frame methods that address a problem setting very similar to ours. PerAct, RVT, and 3D Diffuser all create point clouds using RGBD data captured from four different camera views. This is exactly the pipeline in our method as well. Specifically, PerAct transforms the raw RGBD input into a point cloud and then into a voxel map. RVT constructs the point cloud and then re-projects it onto orthographic images. 3D Diffuser Actor is also conditioned on the entire point cloud. The only difference concerning the input data between our method and these baselines is that we use per-object segmentation masks.

RPDiff [4] is the baseline that consumed the same segmented point cloud ($P_a$ and $P_b$) as ours. It iteratively denoises the randomly sampled relative transformation poses conditioned on the current configuration of $P_a$ and $P_b$. However, it can only solve the single-step place problem trained as a single-task policy. To make a fair comparison, we adapted it to a multi-task key-frame prediction model. Similar to our settings, we consider the pick problem as inferring the relative pose between the gripper ($P_a$) and the object to grasp ($P_b$). The preplace action prediction can also be viewed as calculating the relative pose between the object ($P_b$) and the placement ($P_b$). We train the pick model and the place model separately, which is similar to ours. Since the original RPDiff learns a single-task single-step policy $f\colon (P_a, P_b) \mapsto T_{ab}$, it requires training 18 different models to solve the six tasks in Table 1. We adapt it to learning a multi-task policy conditioned on the language embedding $f\colon (P_a, P_b, f_\ell) \mapsto T_{ab}$. Specifically, we used the same language embedding generated from CLIP [41]. To make RPDiff consume the language embedding, we map it to a 128-dimension feature via a linear layer and concatenate it to a 130-dimension time step embedding (the diffusion step). The model was trained and evaluated with the same settings in [4].

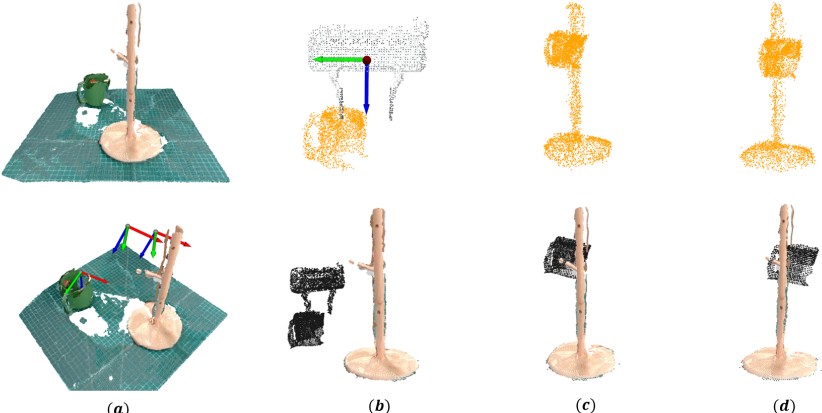

(a)      (b)      (c)      (d)

*Figure 10.* Action inference on *Mug-Tree* with real-sensor data: (a) the observed real-sensor point cloud and the inferred pick, preplace and place action from IMAGINATION POLICY, (b) pick generation, (c) preplace generation, and (d) place generation. The top row shows the generated points with orange color and the bottom row demonstrates the configurations of pick, preplace, and place with the calculated rigid transformations. Please note that we used the point cloud from Franka-Emika Panda gripper to train the model and evaluated it with the Robotiq-85 gripper.

## 6.5    Real-robot Experiments Pipeline

Figure 10 illustrates our pipeline of key-frame action inference on *Mug-Tree* task with real-sensor data. The observed point cloud is shown in the first row of Figure 10a. The predicted pick, preplace and place action from IMAGINATION POLICY are plotted with RGB frames in the second row of Figure 10a. Specifically, Figure 10bcd illustrate the pick generation, preplace generation, and place generation respectively.

For execution on a robot, it requires a collision-free pick-and-place trajectory that connects the key-frame action. We use RRT-star as our motion planner and add the configuration of obstacles (the table, the mounting, and the cameras) to the planner to generate the trajectory.

## 6.6    Detailed Results on RLbench task

We report the results of our method and baselines on RLbench tasks with $\pm 1.98$ std error in Table 6.

| Model | # demos | phone-on-base | stack-wine | put-plate | put-roll | plug-charger | insert-knife |
|---|---|---|---|---|---|---|---|
| IMAGINATION POLICY (ours) | 1 | $4.00 \pm 4.52$ | $2.67 \pm 2.61$ | $1.33 \pm 2.61$ | $2.78 \pm 2.72$ | 0 | 0 |
| IMAGINATION POLICY (ours) | 5 | $78.67 \pm 10.45$ | $\mathbf{97.33 \pm 2.61}$ | 0 | $1.39 \pm 2.71$ | $24.00 \pm 1.57$ | $38.67 \pm 2.61$ |
| IMAGINATION POLICY (ours) | 10 | $\mathbf{90.67 \pm 2.61}$ | $\mathbf{97.33 \pm 2.61}$ | $34.67 \pm 10.45$ | $\mathbf{23.61 \pm 5.44}$ | $\mathbf{26.67 \pm 13.82}$ | $\mathbf{42.67 \pm 9.42}$ |
| RVT [2] | 10 | $56.00 \pm 4.52$ | $18.67 \pm 2.61$ | $\mathbf{53.33 \pm 6.91}$ | 0 | 0 | $8.00 \pm 4.52$ |
| PerAct [1] | 10 | $66.67 \pm 11.39$ | $5.33 \pm 2.62$ | $12.00 \pm 4.52$ | 0 | 0 | 0 |
| Diffusor 3D [3] | 10 | $29.33 \pm 5.22$ | $26.67 \pm 14.55$ | $12.00 \pm 0$ | 0 | 0 | 0 |
| RPDiff [4] | 10 | $62.67 \pm 5.22$ | $32.00 \pm 4.52$ | $5.33 \pm 5.22$ | 0 | 0 | $2.67 \pm 2.61$ |
| Discrete Expert | | 98.67 | 100 | 74.6 | 56 | 72 | 90.6 |

*Table 6.* Detailed performance comparisons on RL benchmark. Success rate (%) on 25 tests v.s. the number of demonstration episodes (1, 5, 10) used in training. Results are averaged over 3 runs. Even with only 5 demos, our method can outperform existing baselines by a significant margin.

