# OpenReview forum: "IMAGINATION POLICY: Using Generative Point Cloud Models for Learning Manipulation Policies"
_robot-learning.org/CoRL/2024/Conference — CoRL 2024_

### Official Review · Reviewer_1BKy · 2024-07-20
**Good pipeline but handles simpler problem**

**Originality:** 2
**Technical Quality:** 4
**Clarity Of Presentation:** 4
**Potential Impact:** 2
**Recommendation:** 3
**Confidence:** 4

**Review:**

This paper proposes to predict the target point cloud from the initial point cloud and language instruction. The method validates broad manipulation tasks and achieves good results.

Although the task benchmark is extensive, the subproblem this work solving only requires rigid transformation. It is possible to use a 6D grasp pipeline to predict the transformation, for example, by sampling 6D poses, and evaluating its quality according to the target point cloud [1].

This work only considers segmented point clouds and does not consider collision-free valid targets.

This pipeline has the potential to generalize to more challenging tasks, for example, dealing with collision [2] or soft body [3].

I think this work requires additional efforts to demonstrate its ability.

[1] A billion ways to grasp: An evaluation of grasp sampling schemes on a dense, physics-based grasp data set

[2] SE(3)-DiffusionFields: Learning smooth cost functions for joint grasp and motion optimization through diffusion

[3] Any-point Trajectory Modeling for Policy Learning

**Quality Of The Limitations Section:**

3

**Questions For Rebuttal:**

1. Is it possible to directly predict two objects' SE(3) poses? Is it possible for a sample-based grasp model to predict the 6D pose?

2. Is it possible to extend this work to consider collision or generate multiple targets?

**Robotics Focus:**

4

**Summary Of Paper:**

This work proposes to use diffusion to generate target point cloud of two objects, and predicts tranformation of objects for manipulation.

**Summary Of Recommendation:**

The paper successfully predicts where objects should go based on initial point clouds and instructions, showing good results with extensive experiments. However, it is possible to use simpler method to solve the task, and current method doesn’t account for more complex situations like avoiding collisions.

---

### Official Review · Reviewer_U3RP · 2024-07-21

**Originality:** 3
**Technical Quality:** 3
**Clarity Of Presentation:** 4
**Potential Impact:** 2
**Recommendation:** 3
**Confidence:** 4

**Review:**

Strengths:

* The idea to use flow models for predicting target point clouds is interesting and novel.
* The accompanying figures and videos are useful in illustrating the ideas introduced in the paper.

Issues:

* The flow of the introduction section feels unnatural. The first two paragraphs motivate the need for local geometric information in robot manipulation policies and introduce Imagination Policy method, but the third paragraph jumps abruptly to discussing about learning from multimodal robot data and dives into related works in that aspect. I suggest editing the intro to improve flow and clarity.
* The authors propose a method that works only with tasks that can be formulated as a three-step-process: pick, pre-place, place. Can the method solve long-horizon tasks that require multiple steps to complete? Can the method deal with manipulation of articulated objects? If the proposed method can solve these tasks, it would be great if the authors can add simulated and real robot experiments that involve these kinds of tasks. If not, the authors should explicitly acknowledge these limitations both in the introduction and the dedicated limitation section. I would be careful with the wording in "state-of-the-art performance across various tasks... compared with several strong baselines" (L10).
* Unlike the baselines that the authors compared with in the paper (RVT, PerAct), the proposed method assumes knowledge of which objects in the scene are ones to be picked up and placed on. This makes the proposed method less general-purpose than the competing baselines.
* The problem formulation in the paper resembles [1,2] more than the baselines mentioned in Table 1 of the paper. Could the authors add a comparison with one of these works in Table 1?
* The authors should move their real robot experiments from the appendix to the main paper. In addition to showing qualitative experiments, it would be great it the authors can add quantitative comparison to at least one baseline and showcase the generalizability of the proposed method when input object appearances and poses change.

Minor corrections:

* Consider citing [2,3] related to equivariant robot manipulation.
* I found that the large amount of symbols in the method section make it difficult to read through and understand. The authors should consider improving or adding figures in the paper and connect the symbols in the text with the figures.

Citations:

1. Simeonov et al. SE(3)-equivariant Object Representations for Manipulation. ICRA 2022.
2. Simeonov et al. SE(3)-Equivariant Relational Rearrangement with Neural Descriptor Fields. CoRL 2022.
3. Yang et al. EquivAct: SIM(3)-Equivariant Visuomotor Policies beyond Rigid Object Manipulation. ICRA 2023.

**Quality Of The Limitations Section:**

2

**Questions For Rebuttal:**

See above.

**Robotics Focus:**

4

**Summary Of Paper:**

This paper proposes Imagination Policy, a method for manipulation policy learning that uses generative flow models to synthesize target points clouds of pick and place motions. For the place phase, the method takes point clouds of the picked object and the receptacle as input, encodes them into SO(3)-invariant point features, generates a joint point cloud containing the two objects in the target pose, and estimates the rigid transformations for both objects between initials poses before and after placing. For the pick phase, the method takes point clouds of the end-effector and the object to be picked up, and perform similar operations, except that the end-effector point cloud is fixed during generation. The method realizes encoder equivariance through extensive SO(3) data augmentation.

**Summary Of Recommendation:**

This paper introduces an interesting idea that suffers from limitations that prevent the proposed method from being used as general-purpose manipulation policies. The clarity of writing needs improvement and the authors need to perform more experiments to make a convincing argument. I recommend the authors spend more time to improve this submission.

---

### Official Review · Reviewer_1GT1 · 2024-07-24
**Pointcloud Diffusion as a Pick-Place action generator**

**Originality:** 5
**Technical Quality:** 1
**Clarity Of Presentation:** 4
**Potential Impact:** 1
**Recommendation:** 2
**Confidence:** 5

**Review:**

The work is very original. The common approach to solve pick and place problems is by learning a model that generates directly the desired rigid transform on the robot (for picking tasks) or on the object (for placing tasks). Instead, the authors propose a novel approach that instead generates the desired point cloud and then, estimate the rigid body by SVD.

**Strenghts**
- The presented idea is very novel and different with respect to the common approaches. If the authors are able to validate experimentally their proposed approach, the method would be of high interest for the robotics community.

**Weaknesses**
- The selected experimetal evaluation is not properly selected. The authors compare their method against PerAct, RVT and 3D Diffuser Actor. Nevertheless, these methods are usually trained asuming raw RGBD data, while their method assume segmented full pointclouds of the interested objects. Thus, the comparison is unfair and unproper to validate the benefit of their method.
- Instead, the authors could compare the benefit of generating a point cloud and infering the rigid body against works that are more similar to theirs. For example, the authors could compare the placing against [1], that consider two pointclouds and instead applies diffusion in the SE(3) space. For the picking, authors could compare against [2], that consider also a target object pointcloud and diffuses the SE(3) pose of the gripper pose. This methods fit better as baselines as they consider the same information available for generating their actions and they only differ in the proposed method in the way the rigid body is estimated.

[1] Simeonov, Anthony, et al. "Shelving, Stacking, Hanging: Relational Pose Diffusion for Multi-modal Rearrangement." 7th Annual Conference on Robot Learning.

[2] Urain, Julen, et al. "Se (3)-diffusionfields: Learning smooth cost functions for joint grasp and motion optimization through diffusion." 2023 IEEE International Conference on Robotics and Automation (ICRA). IEEE, 2023.

**Quality Of The Limitations Section:**

1

**Questions For Rebuttal:**

- Given the originality and novelty of their approach, the authors should prove experimentally that the proposed approach to generate rigid bodies (pointcloud diffusion + SVD) is better (or at least to the level) than directly generating rigid body (SE(3) diffusion).

**Robotics Focus:**

3

**Summary Of Paper:**

The authors introduce a novel approach to generate pick and place actions. The proposed method receives as input two pointclouds and diffuse them to generate a desired relative configuration between them. Then, the rigid body transforms are recovered through SVD.

**Summary Of Recommendation:**

The method is very original, but the selected experiments are not properly selected to evaluate the benefit of their proposed method. Then, it is hard to validate if the idea is useful for solving robotics problems.

---

### Author Rebuttal · Authors · 2024-08-06

## General Response:
Thanks everyone for your feedback. There are a couple of key points that we want to clarify.

1. **Our use of segmentation masks.** Multiple reviewers commented our comparisons with PerAct, RVT and 3D Diffuser are not fair because we assume the input point cloud can be segmented into distinct objects. This is true. However, there are no methods that exactly match our assumptions (point clouds with object segmentations, multi-task, key-frame prediction, etc.). At the request of the reviewers, we are adding a new comparison to [1], although this method does not match our setting exactly either. Fundamentally, we believe that the SOTA in object segmentation (SAM [4] and SAM-v2 [5]) has become so good that the segmentation assumption is reasonable.

2. **Baseline selection.** We argue that the closest standard methods are PerAct, RVT, and 3D Diffuser. These are multi-task key-frame methods that address a problem setting similar to ours. The reviewers have suggested we baseline against [1][2][3]. We are adding [1] as a new baseline. Nevertheless, we think [1][2][3] are not great baselines for the following reasons. 1). They can only infer a one-step action for a single task whereas our method infers both pick and place actions for multiple tasks; 2). NDFs [2][3] require significant per-object pretraining whereas our method does not.

3. **Baseline comparison.** There was also concern that our baselines were not good choices because they take raw RGBD data as input rather than point cloud input as our method does. Please note that all baselines use three RGBD cameras (as we do) and convert the data to point cloud format before doing any additional processing. PerAct transforms the point cloud into a voxel map. RVT re-projects it onto orthographic images. 3D Diffuser is also conditioned on the entire point cloud.

4. **Strength of our method.** Modulo our use of segmentation masks and the discussion above, our method outperforms PerAct, RVT, 3D Diffuser on several RLBench tasks significantly (Table 1). Moreover, the approach is very novel wrt those prior works. Also, the work is validated on a physical robot. We feel this is a strong paper.

[1] Shelving, Stacking, Hanging: Relational Pose Diffusion for Multi-modal Rearrangement.

[2] SE(3)-equivariant Object Representations for Manipulation.

[3] SE(3)-Equivariant Relational Rearrangement with Neural Descriptor Fields.

[4] Segment Anything.

[5] SAM 2: Segment Anything in Images and Videos.

---

### Decision · Program_Chairs · 2024-09-04

**Decision:**

Accept

**Comment:**

Strengths:
+ Novel idea of using generative point-cloud models and flow models for manipulation.
+ The experimental setup includes evaluations on several simulated manipulation tasks.
+ The figures in the paper are informative.

Weaknesses:
- The approach assumes segmented point-clouds of objects, whereas the baselines – RVT, PerAct, 3D Diffusor Actor – do not make this assumption. All reviewers highlighted that this difference makes the comparison unfair.
- The formulation is limited to three-step-process tasks: pick, pre-place, and place. It is also limited to rigid-body transformations and does not reason about collisions.
- The writing can be greatly improved. The flow of the introduction is abrupt. Real-robot experiments are in the appendix.
- Evaluations are missing comparisons to SE(3)-equivariant approaches such as Simeonov et al 2023, 2022.

Post Rebuttal:
The authors addressed majority of the concerns and added another baseline with segmented input.